# Effects of Three Types of Polymeric Proanthocyanidins on Physicochemical and In Vitro Digestive Properties of Potato Starch

**DOI:** 10.3390/foods10061394

**Published:** 2021-06-16

**Authors:** Jiahui Xu, Taotao Dai, Jun Chen, Xuemei He, Xixiang Shuai, Chengmei Liu, Ti Li

**Affiliations:** 1State Key Laboratory of Food Science and Technology, Nanchang University, Nanchang 330047, China; xl19961019@126.com (J.X.); chen-jun1986@hotmail.com (J.C.); shuaixixiang1989@163.com (X.S.); liuchengmei@aliyun.com (C.L.); 2Agro-Products Processing Science and Technology Research Institute, Guangxi Academy of Agricultural Sciences, Nanning 530007, China; daitaotao@gxaas.net (T.D.); xuemeihe1981@126.com (X.H.)

**Keywords:** polymeric proanthocyanidin, degree of polymerization, starch, pasting, retrogradation, digestibility

## Abstract

The effects of three types of polymeric proanthocyanidins (PPC) with different degrees of polymerization (DP), namely PPC1 (DP = 6.39 ± 0.13), PPC2 (DP = 8.21 ± 0.76), and PPC3 (DP = 9.92 ± 0.21), on the physicochemical characteristics and in vitro starch digestibility of potato starch were studied. PPC addition (5%, *w/w*) increased the gelatinization temperature and decreased some viscosity indices of potato starch, including the peak, trough, breakdown, and setback viscosities. Starch-PPC pastes showed reduced thixotropy and improved stability and gelling properties compared to starch paste. The three types of proanthocyanidins all showed evident inhibitory effects on the digestion and retrogradation of potato starch, including short-term and long-term retrogradation. Among the three, PPC with a lower DP had stronger effects on the starch short-term retrogradation and gelling performance, whereas larger PPC molecules exhibited a greater impact on starch recrystallization and digestive characteristics. The research consequences were conducive to explore the application of functional PPC in starch-based food processing.

## 1. Introduction

Potato starch (PoS) has attracted extensive attention in regard to its physicochemical characteristics, such as digestibility, fine textural properties, high swelling power, and transparency. In addition to producing sacchariferous products, potato starch not only plays a unique role in processing flour products, snacks, and chemical starch, but it also serves as a food additive, including gelling agents and film-forming agents used in dairy products, baked foods, beverages, and other foods [1,2]. However, the application of native potato starch has sometimes been limited due to its thermal instability, easy retrogradation, and high content of rapidly digesting starch. Mixing potato starch with polyphenols has been a research hotspot, because polyphenols could play an important role in enhancing starch performance in different applications [3,4,5,6].

Polymeric proanthocyanidins (PPC) are a kind of polyphenol widely distributed in the plant realm [7]. These compounds are complicated polymers of flavan-3-ol monomer units, with a degree of polymerization (DP) that is generally more than five [8]. Evidence suggests that they are more efficient in the regulation of starch digestion compared to monomeric polyphenols. Li et al. [9] and Mkandawire et al. [10] indicated a positive relationship between the DP value (DP ≥ 5) of proanthocyanidin and the inhibition of α-amylase and α-glucosidase, accordingly modifying starch intestinal glucose release. Barros et al. [11] found that proanthocyanidin-rich sorghum extracts elevated the content of resistant starch in corn starch to nearly three times that of monomer polyphenols. Extensive research has shown that physicochemical properties and biological activities of proanthocyanidins depend on the DP. The physical conformation of high DP proanthocyanidins provides more hydrophobic sites, while lower DP counterparts exhibit a stronger water-soluble capacity due to their considerable external hydroxyl groups [12]. Besides this, Arimboor and Arumughan [13] and Dai et al. [14] discovered that smaller-molecular proanthocyanidins had stronger antioxidant effects. Khadri et al. [15] illustrated that the anti-inflammatory activity of lemon grass proanthocyanidin relied on the DP. Pierini et al. [16] and Kawahara et al. [17] found that proanthocyanidin with a higher DP achieved a stronger growth inhibition influence on some specific cancer cells (e.g., hepatoma carcinoma, esophageal adenocarcinoma, and colon cancer cells).

To date, several papers have focused on the influence of proanthocyanidin on pasting, thermal, rheological, and digestion characteristics of potato starch. Zhang et al. [18] found that grape seed proanthocyanidins increased thermostability, reduced rapid digestion, and decreased the final viscosity and hardness of PoS gel significantly. Gao et al. [19] reported that oligomeric procyanidins (DP ≈ 2.88) improved the viscoelasticity and restrained the retrogradation of PoS paste. However, these studies concentrated on the influence of proanthocyanidins of a certain DP on the potato starch properties, while there have been few studies that have investigated the association between the DP value and the effect, especially for polymeric proanthocyanidin with different DPs. Proanthocyanidins with different degrees of polymerization might exert varied effects on the characteristics of potato starch. Thus, this research provides a new perspective for the influences of DP-sensitive polymeric proanthocyanidin with different DPs on the physical and chemical characteristics and in vitro starch digestibility of potato starch. It is likely to supply a theoretical basis for quality improvement of PoS-based foodstuffs.

## 2. Materials and Methods

### 2.1. Materials

Polymeric proanthocyanidins (PPC, purity ≥ 95%) derived from grape seed were purchased from Solarbio Technology Co., Ltd. (Beijing, China). Potato starch (PoS) was obtained from Rogate Starch Co., Ltd. (Jiangsu, China), and the contents of lipid, water, ash, and amylose were 0.32%, 9.86%, 0.25%, and 26.61% (*w/w*), respectively. All other chemicals were of reagent grade and supplied by Aladdin Co., Ltd. (Shanghai, China).

### 2.2. Purification and Fractionation of PPC by a Sephadex LH-20 Column

The PPC was fractionated with a Sephadex LH-20 column, according to a means expounded in the literature, with certain modifications [8,19]. A total of 6 g of proanthocyanidin sample was dissolved in a methanol/water solution with the volume ratio of 1:1 and loaded in a Sephadex LH-20 column that was pretreated. Then, the pillar was washed with 50% methanol until the eluent became colorless to clear other phenolic substances, glycosides, and some flavan-3-ol monomers. The column was then developed with the sequence of acetone/methanol/water solutions (respectively, 2:5:3, 4:3:3, and 7:0:3 *v/v/v*), and the volume of all solvent mixtures was 600 mL. The grading elutions were named as fraction 1, 2, and 3, respectively. Then, the organic solvents of these fractions were removed by vacuum rotary evaporation and lyophilized to drying. The obtained components were, respectively, labeled PPC1, PPC2, and PPC3.

### 2.3. Determination of the Degree of Polymerization of Proanthocyanidin

The average degrees of polymerization of PPC1, PPC2, and PPC3 were determined by the vanillin-hydrochloric acid method [20,21]. Briefly, the standard curves of catechin’s mass concentration and molar concentration were drawn in sequence (the standard curve of catechin mass concentration was y = 2020.6x + 0.6963, R^2^ = 0.9954; the standard curve of catechin molar concentration was y = 0.1756x − 0.0013, R^2^ = 0.9999). A total of 10 mg of the proanthocyanidin sample thoroughly dissolved in 20 mL methanol solution, which was then diluted 20 times with methanol and acetic acid and, respectively, named as liquid A and liquid B. One milliliter of liquid A and liquid B was applied to measure the content (*m*) and the amount of substance (*n*) of proanthocyanidin according to the methods of standard curves of catechin mass concentration and catechin molar concentration, respectively [22]. In this research, the value of DP was computed by the formula below (1):(1)$$DP=\frac{m}{M\times n}\;$$
where *m* refers to the proanthocyanidin content, *n* is the amount of substance of proanthocyanidin, and *M* is the relative molecular mass of catechin.

### 2.4. Rapid Viscosity Analysis (RVA)

The pasting characteristics of PoS and PoS-PPC blends were determined using RVA (Perten RVA 4500, Stockholm, Sweden), according to preceding papers [18]. Based on our previous research, we chose 20:1 as the mass ratio of starch to PPC [18]. Briefly, 2 g of starch was mixed with proanthocyanidins (PPC1, PPC2, and PPC3) at doses of 0% and 5% (*w/w*), and added with 20 mL of distilled water. The slurries of PoS and PoS-PPC samples were characterized using the RVA, in accordance with the description of Li et al. [23]. The preliminary stirring speed was 960 r/min; then, it was kept at 160 r/min after 10s. The sample was maintained at 50 °C for 90s, heated and maintained at 95 °C for 150s, and decreased and kept at 50 °C for 90s. The pasting curves and parameters of starch pastes were generated by TCW (Thermocline for Windows). Some of the gelatinized samples were transferred for rheological analysis, and others were reserved at 4 °C for a week to prepare the retrograded samples. In addition, the same mass ratio of starch to PPC was used to investigate DSC, XRD, and in vitro digestibility of starch-PPC samples.

### 2.5. Rheological Measurements

The rheological characteristics of PoS and PoS-PPC compounds were analyzed by an MCR 302 rheometer (Anton-Paar, Graz, Austria) with a parallel plate measuring system, according to Kong et al. [24] and Chen et al. [4]. The gels of starch and starch-PPC mixtures obtained from Section 2.4 were transferred to a rheometer board with a probe type of PP50 and a gap of 0.5 mm. The pastes were equilibrated at room temperature for 5 min before measurement.

#### 2.5.1. Steady Shear Analysis

The changes in shear stress of the samples were determined within the range of increasing shear rate from 0–1000 s^−^^1^ and then decreasing from 1000–0 s^−1^. The obtained curve was fitted with the Power Law equation for regression fitting:(2)$$\sigma =K\;\cdot {\;\gamma }^{n}\;$$
where *σ* is the shear stress (Pa). *γ* is the shear rate (s^−1^). *K* is the consistency coefficient (Pa·s^n^). *n* refers to the flow behavior index (*n* < 1 for pseudoplasticity fluid, and *n* = 1 for Newtonian fluid).

#### 2.5.2. Dynamic Rheological Analysis

An oscillatory frequency sweep mode (0.2–20 Hz) was conducted at room temperature with 1% strain. The storage modulus (G′) and loss modulus (G″) were obtained.

### 2.6. Differential Scanning Calorimetry (DSC)

The thermal characteristics of PoS and PoS-PPC samples were measured with DSC (7000X, HITACHI, Japan) on the basis of He et al. [25] and Xu et al. [26]. Some parameters were achieved from the DSC curves, including the onset (T_o_), peak (T_p_), conclusion temperature (T_c_), gelatinization (Δ*H_g_*), and retrogradation enthalpy (Δ*H_r_*). Finally, the retrogradation degree *R* (Δ*H_r_*/Δ*H_g_*) was computed on account of previous research [27].

### 2.7. X-ray Diffraction (XRD)

XRD (D8 Advance, Bruker, Karlsruhe, Germany) was used to determine the crystallinity of PoS and PoS-PPC complexes, based on a previous paper [28]. The scanning range of the diffraction angle (2θ) was 4°–40°. The relative crystallinity was computed by an integral using the Origin software (3):(3)$$Relative\;crystallinity\;(\%)=\frac{A_{c}}{A_{c}{+A}_{a}}\times 100\;$$
here *A_c_* and *A_a_* stand for the crystal and amorphous areas, respectively.

### 2.8. In Vitro Digestion

The in vitro digestibility of PoS and PoS-PPC compounds was analyzed in accordance with the preceding literature, with some modifications [29,30]. A total of 200 mg of starch or starch-PPC mixture samples was dissolved in 5 mL water, fully gelatinized, cooled, and added with 15 mL phosphate buffered solution (pH = 5.2). Subsequently, 5 mL of mixed digestive enzyme fluid (13 U/mg porcine pancreatic α-amylase, 260 U/mg glucoamylase) was added to the PoS or PoS-PPC mixtures, and then incubated at 37 °C. Other steps refer to previous studies. The contents of rapidly digested starch (RDS), slowly digested starch (SDS), and resistant starch (RS) in starch fractions were computed with the Equations (4)–(6) below:(4)$$RDS\;(\%)=\frac{G20-FG}{TG}\times 0.9\times 100\;$$
(5)$$SDS\;(\%)=\frac{G120-G20}{TG}\times 0.9\times 100\;$$
(6)$$RS\;(\%)=\frac{TG-\left(RDS+SDS\right)}{TG}\times 100\;$$

Here, *G*20 and *G*120 refer to the contents of glucose generated at 20 and 120 min of the hydrolysis process, respectively. *FG* is the content of free glucose in samples, and *TG* is the total amount of starch in the original sample. 

### 2.9. Statistical Analyses 

The data were represented as the means ± standard deviation (SD) conducted in triplicate analyses. Duncan’s test was carried out in SPSS 24.0 statistical software, and at *p* < 0.05, the significant difference was set.

## 3. Results and Discussion

### 3.1. Qualitative Analysis of PPC

On the basis of the difference in the molecular weight, molecular structure, and molecular polarity of proanthocyanidins with different DPs, we reported on the use of such a gradient with acetone/methanol/water solutions (respectively, 2:5:3, 4:3:3, and 7:0:3 *v/v/v*) to preparatively separate the proanthocyanidins [9,31]. With the increase of acetone volume fraction, the average degrees of polymerization of the fractionated proanthocyanidins increased successively, which were 6.39 ± 0.13, 8.21 ± 0.76, and 9.92 ± 0.21, respectively.

### 3.2. Pasting Properties

Table 1 and Figure 1 exhibit the pasting properties of PoS in the presence of PPC with three DPs. As shown in Figure 1, with the temperature increasing, starch granules began to swell under the action of water, the crystalline region was destroyed, and amylose subsequently diffused and leached out [27]. Visually, all of the viscosity curves increased markedly but showed an obvious difference between PoS and PoS-PPC mixtures, with 3491, 2397, 2736, and 2591 mPa·s for PoS, PoS-PPC1, PoS-PPC2, and PoS-PPC3, respectively. The addition of PPC1, PPC2, and PPC3 led to significant decreases in peak viscosity (PV) and increases of pasting temperature (PT) of PoS from 68.63 to 68.66 and 69.00 and 69.53 °C, respectively, which indicates that PPC delayed the rapid swelling of starch granules. Similar results were reported in previous works [18], which found that the released polyphenols crosslinked starch with water through polyphenolic hydroxyl groups. On the one hand, PPC diminished water usability for swelling of starch particles via competition for water and then decreased the amount of leaching amyloses and overall viscosity. On the other hand, the connection between PPC and starch chains, especially amylose, weakened the direct interaction between starch granules and moisture [32]. The newly formed starch-PPC complex might have wrapped onto the surface of starch particles and further inhibited the hydration of amorphous regions and pasting process of starch. Moreover, the negative charges of phosphate monoesters in PoS repelled each other, which promoted the rapid swelling of PoS granules. However, interactions between phosphate monoesters of starch and hydroxy groups of proanthocyanidin might happen during hydration, thus retarding the blends’ gelatinization [33].

Among the three types of PPCs, PPC1 had the strongest effect on the pasting viscosity. This could be explained by the hygroscopicity of proanthocyanidins varying with their DP. PPCs with a slightly lower DP exhibited stronger moisture-absorption capacity because of a lot of external hydroxyl groups, thus reducing the exposure of starch molecules to water [19]. Presumably, PPCs of a smaller size could hamper the entanglement and formation of the double helix crystalline region of amylose, resulting in a decrease in starch viscosity. As the DP value increases, proanthocyanidin molecules become larger and their spatial structures get more complicated [34]. Vernhet et al. [35] described that highly condensed proanthocyanidins showed a denser shape with ramifications. To maintain the system stability of PPC molecules, intramolecular hydrophobic interactions and hydrogen bonds between the groups tend to be formed, which lead to a less extended structure and lower solubility. 

In the case of BD, it reflects the stability and integrality of the starch, and a high BD value suggests that granules were easily damaged [33,36,37]. PPC1, PPC2, and PPC3 addition caused decreases of the BD value from 1906 to 902, 1202, and 1061 mPa·s, respectively. The results showed that PPC could improve the thermal and shearing stability of potato starch. The effect was most pronounced for short-chain PPCs with the smallest molecules. This might be attributed to the increased conformational constraints for higher-DP proanthocyanidins, which impeded the interaction between proanthocyanidins and starch molecules to form a network structure within a relatively short time. The SB value reflects the degree of amylose reaggregation and recrystallization during short-term retrogradation [24]. The three types of PPCs could significantly reduce SB values to varying degrees, for PPC1 the strongest and PPC3 the least, from 397 to 248 and 330 mPa·s, respectively. This phenomenon might be because the connection of PPC and amyloses was stronger than the rearrangement between amylose molecules, resulting in a decrease of SB value [24]. In addition, PPC molecules perhaps generated a water layer around starch granules to limit the rearrangement of amylose during the gelatinization process; low-DP proanthocyanidin had a higher water retention capacity and reduced the fluidity of starch chain, thereby restraining short-term retrogradation of starch.

### 3.3. Rheological Measurements

#### 3.3.1. Steady Shear Analysis

The flow characteristic curves of the PoS and PoS-PPC complexes are shown in Figure 2, and all the fitting parameters are listed in Table 2. Since the correlation between shear stress and shear rate was nonlinear and all *n* values were less than 1, the PoS and PoS-PPC blends were non-Newtonian fluids, exhibiting a shear-thinning property (a pseudoplasticity behavior). As for all the samples, stress shear hysteresis loops appeared in the shearing process, indicating a thixotropic effect. A large hysteresis ring area signifies a strong thixotropic property [38]. The *K* value is related to the starch viscosity, and the larger the *K* value, the poorer the system mobility and shear stability [39]. The hysteresis ring area of starch and consistency coefficient, *K,* decreased with the addition of PPC, which suggested that the starch paste thixotropy was reduced and stability was improved by adding PPC, and the effect was more noteworthy for PPC with a lower DP. This finding was in agreement with the BD result of RVA. The same trends of *K* value and hysteresis ring area of starch added with polysaccharides have been reported before, and these results indicated that the relative stability and uniformity of the starch gel structure was enhanced [4,19]. 

The value of *n* is inversely correlated with the pseudoplasticity degree of the entire system [31], PPC replacement decreasing the *n* value, for PPC1 the most. This meant that the presence of PPC made the entangled starch molecular chains have a greater tendency to be straightened or dispersed under the action of the external force, which was conducive to the flow of starch paste, consequently lowering the viscosity more obviously. The result was unanimous to the change of the RVA test.

#### 3.3.2. Dynamic Rheological Analysis

The viscoelastic properties were determined by dynamic rheological analysis (Figure 3). The storage modulus (G′) represents the starch elastic properties and reflects the ability of the material to restore its original state after deformation, while the loss modulus (G″) represents the viscous nature of starch colloid and characterizes the ability of the material to resist flowing [4]. G′ and G″ of all samples showed mild frequency dependence; in the total frequency range, the value of G′ was markedly higher than the G″ value, indicating that PoS and PoS-PPC blends displayed weak gel-like behavior [40]. Replacement of PoS with PPC increased the mechanical modulus, especially G′. This result indicated that PPC prominently affected the elastic properties instead of the viscous characteristics of the PoS-PPC mixtures. As a higher G′ means greater rigidity and strength of the PoS paste network structure, proancyanidins with a slightly lower DP could strengthen the gel network of starch more significantly [41]. Because amylose is the main polymer that forms the cross-linked network [42], we can speculate that the connection of PPC and starch internal amyloses was enhanced through hydrogen bonds, resulting in the increase of the elastic property of starch paste. The more externally active hydroxyl groups of PPC1 increase the entanglement points between molecular chains, facilitating the formation of a three-dimensional gel network structure [40].

In summary, starch-PPC pastes had better gelling properties than starch paste, and the lower the degree of PPC polymerization, the more obvious the effect. PPC addition could make PoS more suitable candidates as gelling agents for food industry manufacturing, including soft candies, ice cream, and meat foods production, thereby improving these products’ nutrition, texture, and stability [43].

### 3.4. Thermodynamic Properties

The thermodynamic parameters obtained from DSC are shown in Table 3. The PPC addition increased the pasting transition temperatures (T_o_, T_p_, and T_c_) of PoS within 2 °C, indicating that PPC could delay starch gelatinization [28]. The results were supposedly attributed to the partial undissolved PPC attached to the starch granules, which enlarged steric hindrance and suppressed the swelling and fracture of starch. Similar effects have been observed by Xiao et al. [44] and Zheng et al. [30], who found that black tea extract and proanthocyanidins from Chinese berry leaves discernibly increased the gelatinization temperatures of rice starch, and they ascribed it to the stability of interactions between these polyphenols and starch.

The gelatinization enthalpy value Δ*H_g_* represents the energy required for the melting of the starch crystallization zone, especially for the double helix structure of amylopectin crystals [45]. Compared to the control (13.50 J/g), the Δ*H_g_* for starch gelatinized with 5% PPC1, PPC2, and PPC3 decreased to 12.17, 8.70, and 8.07 J/g, respectively. PPC has a polyhydroxyl structure. The OH groups of PPC interacted with the side chain of amylopectin and bound to the amorphous area of starch particles to varying degrees. They facilitated the transition from the water-absorbing swelling in the amorphous zone to the crystalline zone for the starch granules, thereby altering the coupling force between the crystalline and amorphous matrix [46]. The consequent hydration easily led to a decrease in gelatinization energy.

The Δ*H_r_* and *R* value of starch mainly reflect the recrystallization level of amylopectin and the long-term retrogradation degree of starch [40]. With the addition of PPC, the Δ*H_r_* and *R* values of starch declined significantly, indicating that the degree of order and crystallinity of starch decreased during the retrogradation process. On the one hand, the strong hydrophilicity of PPC reduced the content of free or available moisture in the starch-PPC system. On the other hand, PPC associated with starch molecules by intense hydrogen bonding and hydrophobicity interactions and, consequently, played a role in steric exclusion, which all limited the activity of the starch chains and retarded retrogradation [47]. Among the three types of PPC, PPC3 had the strongest suppression effect on starch retrogradation, then PPC2, and finally, PPC1. This might be ascribed to the increased binding affinity and specificity for larger proanthocyanidins with a multidentate character [48], which also showed that macromolecular proanthocyanidins possibly had a greater impact on amylopectin and could better inhibit amylopectin recrystallization.

### 3.5. X-ray Diffraction Patterns

The results of XRD are shown in Figure 4. Native potato starch (NPoS) displayed a typical B-type crystal structure, with representative peaks appearing at 2θ = 5°, 15°, 17°, 21°, and 24°. During gelatinization, the crystalline structure of starch granules was destroyed by hydrothermal treatment and converted into an amorphous form [33]. However, during retrogradation, the starch chains tended to recombine into an ordered crystal structure and show a B-type XRD spectra (2θ ≈ 17°), which is principally due to rearrangement of the amylopectin part [6,49]. The supplementation of PPC led to the disappearance of starch B-type diffraction peaks, indicating that PPC could effectively restrain the recrystallization of amylopectin. Studies have shown that starch will form V-type crystals when combined with fatty acids, iodine, and other substances, and the V-type inclusion shows representative diffraction peaks at 20.0° and sometimes at 7.0° and 13.0° [50]. The complex had an obvious diffraction peak at 2θ = 13.1°, which was primarily due to the formation of amylose–lipid complexes, further hindering amylose rearrangement [28].

After adding PPC1, PPC2, and PPC3, the relative crystallinity of potato starch decreased from 17.94% to 16.86%, 15.54%, and 14.44%, respectively. It is speculated that when embedded in the starch-moisture matrix, PPC could have interfered with the rearrangement of starch chains, especially for amylopectins, and a hydrogen-bond interaction might have been the main cause of their mutual effects [51]. The result also showed that larger proanthocyanidin molecules exerted a more significant influence on starch crystallinity than smaller ones, which indicated that proanthocyanidins with high DP values were more likely to destroy the long-range ordered crystalline structure and, thus, retard starch long-term retrogradation. Due to their large steric hindrance, macromolecular polyphenols are difficult to insert into the cavity of the amylose double helix to form inclusion compounds through hydrophobic interactions [52,53]. In other words, long-chain PPCs affect the overall distribution and arrangement instead of short-range double helical construction of the starch chains [30]. This result was consistent with the DSC.

PoS is often served as a substrate for producing starchy packaging films due to its edibility and biodegradability. However, on account of the shortcomings of PoS in processing, starch film production faces the problems of poor mechanical properties and easy crystallization [54,55]. Through our research, we found that PPC as a functional compound can not only serve as a mechanical reinforcement to form an interpenetrating network with starch and enhance the mechanical characteristics of starch films, but it can also suppress the recrystallization and retrogradation of starch, thus maintaining the preserved starch film as flexible and stretchable.

### 3.6. In Vitro Digestibility

As presented in Figure 5, the influences of three types of PPC on the RDS decrease (by 6.06% of PPC1, 9.93% of PPC2, and 13.68% of PPC3) and SDS increase (by 13.41% of PPC1, 21.90% of PPC2, and 20.18% of PPC3) were all significant (*p* < 0.05). PPC3 also showed a prominent effect in the improvement of RS content, with 16.35%. These results indicated that PPC could effectively hinder the digestibility of potato starch, and the inhibition impact was closely associated with the DP of PPC. In general, the replacement of PoS with PPC3 had the strongest suppression of starch digestibility among the three. 

For the following probable reasons, on the one hand, PPC could bind to the amino acid residues and non-competitively inhibit the catalysis of α-amylase and α-glucosidase, mainly through hydrophobic interaction, thus reducing the activity of enzymes [56]. As the DP value of PPC increases, the molecule hydrophobicity tends to be enhanced, which is advantageous to the formation of multiple interaction sites between PPC and amylase [9,11]. Zhou et al. [57] explained that the affinities for α-amylase varies among procyanidins with different DPs, mainly due to their different stabilization on the surface near to the catalytic core of α-amylase. In addition, the low swelling degree was unfavorable to the interaction of enzymes with the starch particles, thereby reducing the content of RDS. On the other hand, much of the literature has showed that starch in the presence of polyphenols might form a nonordered crystal structure with a slow digestion characteristic [5,58]. However, large-molecular-weight proanthocyanidin was likely to provide more hydroxyl groups and hydrophobic domains to favor the interactions with starch molecules. Based on this, by adding candidates of PPC, it was expected to produce natural anti-diabetic drugs or chronically digestive starch that are conductive to postprandial glucose control.

## 4. Conclusions

In this study, we utilized three types of PPC with different DPs (6.39 ± 0.13, 8.21 ± 0.76, and 9.92 ± 0.21, respectively) to research the influence of PPC on physicochemical and in vitro digestion properties for better application of PPC in starchy food. PPC enhanced the thermal stability and delayed starch gelatinization of PoS. It pronouncedly affected the elastic properties, instead of the viscous properties, of PoS pasting. The reduced thixotropy indicated that the relative stability and uniformity of the starch gel structure was enhanced, showing better gelling properties. Starch digestion and retrogradation could be restrained by PPCs with different DPs in varying degrees. Among the three, PPCs with a lower DP more evidently influenced starch gelling performance, whereas larger PPC molecules exhibited a greater impact on starch long-term retrogradation and digestive properties. This might be attributed to the different type of starch chains combined with PPCs of different DP. The findings of the present study may render a beneficial recommendation for the application of PPC to modify gelatinization, thermodynamic, rheological, and digestive characteristics of PoS-based foods.

## Figures and Tables

**Figure 1 foods-10-01394-f001:**
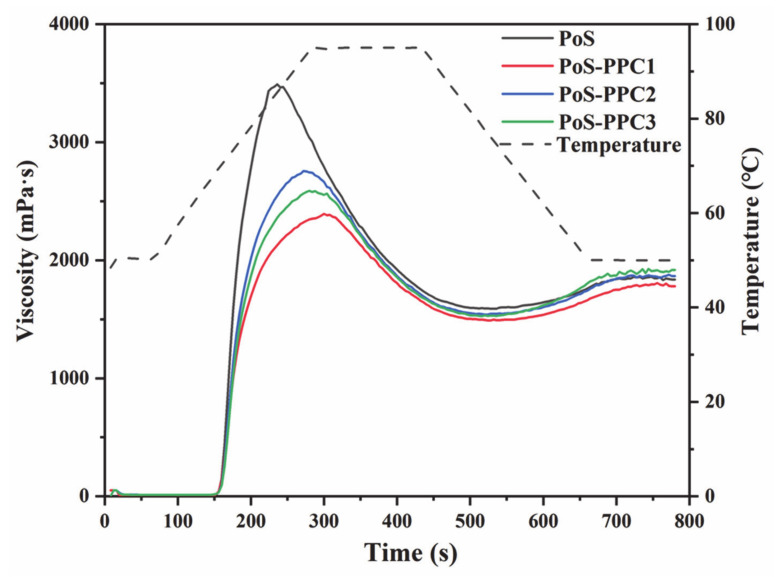
RVA gelatinization curves of native PoS and PoS-PPC pastes. PoS: potato starch; PPC: polymeric proanthocyanidin.

**Figure 2 foods-10-01394-f002:**
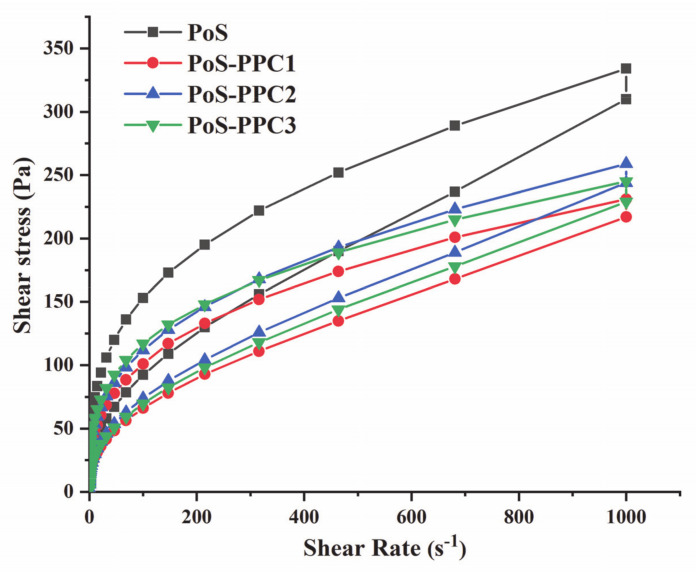
Steady flow curve of PoS and PoS-PPC pastes. PoS: potato starch; PPC: polymeric proanthocyanidin.

**Figure 3 foods-10-01394-f003:**
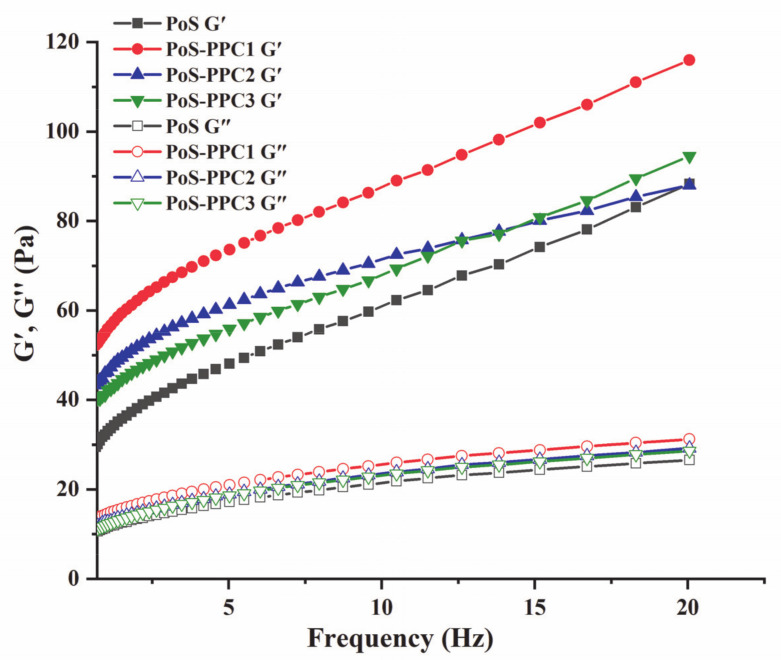
Variation tendency of storage modulus (**G′**) and loss modulus (**G″**) with frequency for PoS and PoS-PPC mixtures. PoS: potato starch; PPC: polymeric proanthocyanidin.

**Figure 4 foods-10-01394-f004:**
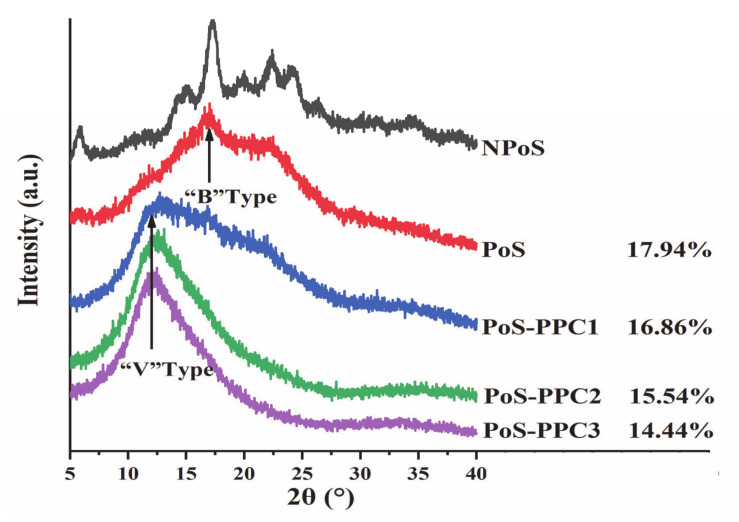
XRD diagrams of PoS and PoS-PPC blends, and the relative crystallinity is shown next to the sample name. NPoS, native potato starch; PoS, potato starch; and PPC, polymeric proanthocyanidin. PoS: potato starch; PPC: polymeric proanthocyanidin; 2θ: Diffraction angle.

**Figure 5 foods-10-01394-f005:**
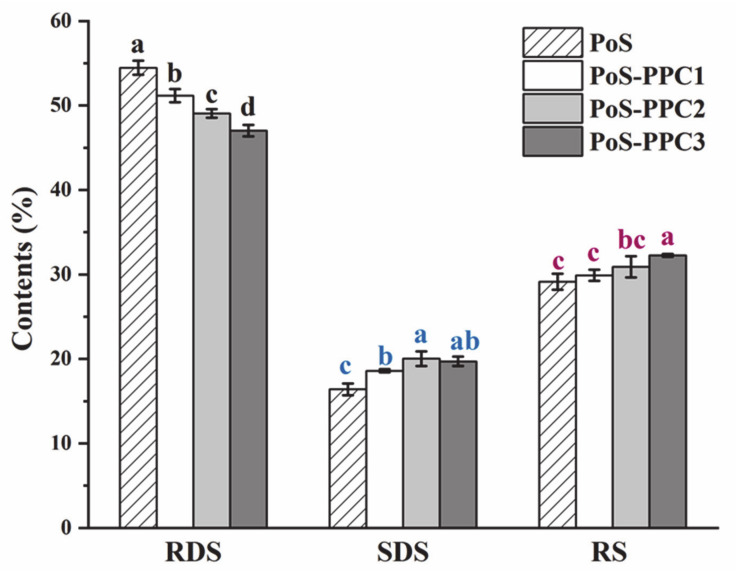
Influence of PPC on in vitro digestibility of potato starch. RDS, rapidly digesting starch; SDS, slowly digesting starch; and RS, resistant starch. PoS, potato starch and PPC, polymeric proanthocyanidin. Different colors of lowercase letters stand for significant differences (*p* < 0.05).

**Table 1 foods-10-01394-t001:** Gelatinization characteristics of PoS and PoS-PPC complexes.

Samples	PV (mPa·s)	TV (mPa·s)	BD (mPa·s)	FV (mPa·s)	SB (mPa·s)	PT (°C)
PoS	3491 ± 1 ^a^	1588 ± 12 ^a^	1906 ± 27 ^a^	1833 ± 13 ^c^	397 ± 8 ^a^	68.63 ± 0.01 ^b^
PoS-PPC1	2397 ± 7 ^d^	1495 ± 6 ^c^	902 ± 1 ^d^	1778 ± 11 ^d^	248 ± 3 ^d^	68.66 ± 0.04 ^b^
PoS-PPC2	2736 ± 20 ^b^	1535 ± 6 ^b^	1202 ± 16 ^b^	1864 ± 4 ^b^	283 ± 6 ^c^	69.00 ± 0.57 ^a^
PoS-PPC3	2591 ± 2 ^c^	1524 ± 4 ^b^	1061 ± 2 ^c^	1919 ± 1 ^a^	330 ± 2 ^b^	69.53 ± 0.04 ^a^

Different letters in a column signify significant differences (*p* < 0.05). PoS: potato starch; PPC: polymeric proanthocyanidin; PV: peak viscosity; TV: trough viscosity; FV: final viscosity; PT: pasting temperature; BD: breakdown; SB: setback.

**Table 2 foods-10-01394-t002:** The steady shear rheological parameters of PoS and PoS-PPC pastes modelled by the Power Law equation.

Samples	Hysteresis Loops Area (Pa·s^−1^)	Up Curve			Down Curve		
		*K*/Pa·s^n^	*n*	*R* ^2^	*K*/Pa·s^n^	*n*	*R* ^2^
PoS	52,972.6	43.55	0.37	0.98	12.11	0.45	0.98
PoS-PPC1	32,590.2	16.81	0.29	0.99	8.93	0.44	0.98
PoS-PPC2	33,943.7	17.82	0.36	0.99	9.30	0.46	0.99
PoS-PPC3	38,943.9	19.56	0.35	0.97	9.05	0.45	0.90

PoS: potato starch; PPC: polymeric proanthocyanidin; *K*: the consistency coefficient; *n*: the flow behavior index.

**Table 3 foods-10-01394-t003:** Thermal properties of PoS and PoS-PPC blends.

Samples	T_o_ (°C)	T_p_ (°C)	T_c_ (°C)	Δ*H_g_* (J/g)	Δ*H_r_* (J/g)	*R* (%)
PoS	60.51 ± 0.29 ^c^	62.43 ± 0.21 ^c^	65.54 ± 0.24 ^b^	13.50 ± 0.12 ^a^	4.54 ± 0.03 ^a^	33.63
PoS-PPC1	61.14 ± 0.01 ^b^	63.16 ± 0.11 ^b^	66.61 ± 0.15 ^a^	12.17 ± 0.13 ^b^	3.13 ± 0.12 ^b^	25.68
PoS-PPC2	61.71 ± 0.05 ^a^	63.63 ± 0.09 ^a^	67.22 ± 0.21 ^a^	8.70 ± 0.19 ^c^	1.93 ± 0.07 ^c^	22.18
PoS-PPC3	61.45± 0.07 ^ab^	63.74 ± 0.08 ^a^	66.84 ± 0.11 ^a^	8.07 ± 0.09 ^d^	1.52 ± 0.08 ^d^	18.83

Different letters in a column signify significant differences (*p* < 0.05). PoS: potato starch; PPC: polymeric proanthocyanidin. T_o_: onset temperature; T_p_: peak temperature; T_c_: conclusion temperature; Δ*H_g_*: gelatinization enthalpy; Δ*H_r_*: retrogradation enthalpy; *R*: the degree of retrogradation (Δ*H_r_*/Δ*H_g_*) × 100.

## Data Availability

Data are contained within the article.

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
