# Peer review of "Effects of Three Types of Polymeric Proanthocyanidins on Physicochemical and In Vitro Digestive Properties of Potato Starch"

_foods, 2021, doi:10.3390/foods10061394_

Round 1
Reviewer 1 Report
The manuscript entitled “Effects of polymeric proanthocyanidin with different degrees of polymerization on physicochemical and in vitro digestive properties of potato starch” reports the effects of three types of polymeric proanthocyanidins with different degrees of polymerization on physicochemical and in vitro digestion properties of potato starch. Overall, the manuscript is well writer and organized. The methods are sufficiently described, with reference to their detailed description, and the results were compared with the literature and may have practical application in the food industry.
Some minor suggestions:
-Line 190. Replace “as described in” with “as shown in”.
-Include the meaning of the abbreviation PoS, PoS-PPC1, etc. in the caption of the different figures.
-Line 247. mixtures are shown in Table…
-In Figure 4, provide units for intensity, if relevant.
Author Response
Response to Reviewer1 Comments
The manuscript entitled “Effects of polymeric proanthocyanidin with different degrees of polymerization on physicochemical and in vitro digestive properties of potato starch” reports the effects of three types of polymeric proanthocyanidins with different degrees of polymerization on physicochemical and in vitro digestion properties of potato starch. Overall, the manuscript is well writer and organized. The methods are sufficiently described, with reference to their detailed description, and the results were compared with the literature and may have practical application in the food industry.
Point 1: Some minor suggestions: Line 190. Replace “as described in” with “as shown in”.
Response 1: Thank you very much for your good suggestion. As suggested, the “as described in” was changed to “as shown in” (Please see page 5, line 194 in the revised manuscript).
Point 2: Include the meaning of the abbreviation PoS, PoS-PPC1, etc. in the caption of the different figures.
Response 2: Thank you for your good suggestions. As suggested, the Figure 1, 2, 3, and 4 have added the meaning of the abbreviation PoS, PPC, etc. in the captions (Please see page 6 (Figure 1), 7 (Figure 2), 8 (Figure 3) and 10 (Figure 4) in the revised manuscript).
Point 3: Line 247. mixtures are shown in Table…
Response 3: Thank you for your good suggestions. We have rephrased this sentence by swapping the positions of “Figure 2” and “Table 2” (Please see page 7, lines 259, 260 in the revised manuscript).
Point 4: In Figure 4, provide units for intensity, if relevant.
Response 4: Thanks. We have added “a.u.” as the unit for intensity in figure 4 (Please see page 10 (Figure 4) in the revised manuscript).
Thank you again. We hope that you could be satisfied with our changes.

Reviewer 2 Report
Very interesting manuscript. Only minor corrections are suggested.
1. Please cite references in text without coma i. e. Name et al.
2. Line 45 - please rephrase this sentence - properties of protoanthocyanidins depend on... not are sensitive
3. have the authors considered the antioxidant properties of modified PoS?
Author Response
Response to Reviewer2 Comments
Very interesting manuscript. Only minor corrections are suggested.
Point 1: Please cite references in text without comma i. e. Name et al.
Response 1: Thank you very much for your good suggestion. We have revised the
references in text without comma i. e. Name et al. (Please see page 1, lines 39, 41 and 42, page 2, lines 48, 49, 51, 56 and 59, page 3, lines 124, and 141, page 6, line 234, page 9, line 317, page 11, line 405 in the revised manuscript).
Point 2: Line 45 - please rephrase this sentence - properties of proanthocyanidins depend on... not are sensitive
Response 2: Thanks. As suggested. We have rephrased this sentence by changing “are sensitive to” to “depend on” (Please see page 1, line 45 in the revised manuscript).
Point 3: have the authors considered the antioxidant properties of modified PoS?
Response 3: Thank you for your good suggestions. We agree with your opinion. It is generally believed that natural PoS does not have antioxidant activity, but starch modified by proanthocyanidins may have certain antioxidant activity. In future studies, it may be useful and significative to examine the antioxidant properties of modified PoS in detail, but this was beyond the scope of the current manuscript.
Thank you again. We hope that you could be satisfied with our changes.

Reviewer 3 Report
The authors used in this study three types of PPC with different DPs (6.39 ± 0.13, 8.21 ± 0.76 and 9.92 ± 0.21, respectively) to investigate the effects of PPC on the physicochemical and in vitro digestion properties for better application of PPC in starchy food
PPC enhanced the thermal stability and delayed starch gelatinization of PoS. It pronouncedly affected the elastic properties more than the viscous properties of this pasting
Starch digestion and retrogradation could be restrained by PPC with different DPs in varying degrees. Among the three, PPC with lower DP more evidently influenced starch gelling performance, whereas larger molecules exhibited a greater impact on starch long-term retrogradation and digestive properties
It might be attributed to the different type of starch chains combined with PPC of different DP. The results of current study may render a beneficial recommendation for the use of PPC to modify pasting, thermal, rheological and digestive properties of PoS-based foods.
Q1 : Title
Title :
Must be shorten and suppress polymeric at least and can be added “for three types of polymeric and resumed as follow :
“Effects of three types of polymeric proanthocyanidin on physiochemical …..
Q2 : Abstract and Keywords
Is OK
Keywords : Are enough in number and well selected for the searching
Q3 . 1. Introduction
Mixing potato starch with polyphenols has been a research hotspot because polyphenols could play an important role in enhancing starch performance in different applications
Polymeric proanthocyanidins (PPC) are a class of polyphenols widely distributed within the plant realm. Evidences suggest that they are more efficient in the regulation of starch digestion compared to monomeric polyphenols.
Extensive researches have shown that physicochemical properties and biological activities of proanthocyanidins are sensitive to DP. The physical conformation of high DP proanthocyanidins provides more hydrophobic sites, while lower DP counterparts exhibit stronger water-soluble capacity due to its considerable external hydroxyl groups
To date, several papers focused on the effect of proanthocyanidin on pasting, ther mal, rheological and digestion properties of potato starch
Proanthocyanidins with different DPs might exert varied influences in the properties of potato starch. Thus, this study provides new insights into the effects of DP-sensitive polymeric proanthocyanidin with different DPs on the physicochemical and in vitro digestive properties of potato starch, supplying a basis for improving the quality of PoS-based foods
Q4.2. Materials and Methods
2.1. Material
Is OK
2.2. Purification and Fractionation of PPC by a Sephadex LH-20 Column
Is OK
2.3. Determination of the degree of polymerization of proanthocyanidin
Is OK
2.4. Rapid viscosity analysis (RVA)
Is OK
2.5. Rheological measurements
Is OK
2.5.1. Steady shear analysis
Is OK
2.5.2. Dynamic rheological analysis
Is OK
2.6. Differential scanning calorimetry (DSC)
Is OK
2.7. X-ray diffraction (XRD)
Is OK
2.8. In vitro digestion
Is OK
2.9. Statistical analyses
Is OK
Q5. 3. Results and Discussion
3.1. Qualitative analysis of PPCV
Is OK
3.2. Pasting properties
As described in Figure 1, with the increase of temperature, starch granules began to swell under the action of water, crystalline region was destroyed, and amylose subsequently diffused and leached out
Among the three types of PPC, PPC1 had the strongest influence on the pasting viscosity. This could be explained by the hygroscopicity of proanthocyanidins varying with their DP. PPC with a slightly lower DP exhibited stronger moisture-absorption capacity due to lots of external hydroxyl groups, thus reducing the exposure of starch molecules to water
The results indicated that PPC improved the stability and integrity of potato starches under heat treatment and mechanical shearing in food processing. The effect was most pronounced for short-chain PPC with the smallest molecules.
Three types of PPC could significantly reduce SB values to varying degrees, for PPC1 the strongest and PPC3 the least, from 397 to 248 and 330 mPa·s, respectively.
3.3. Rheological measurements
3.3.1. Steady shear analysis
The relationship between shear stress and shear rate was nonlinear and n was all less than 1, the PoS and PoS-PPC blends were non-Newtonian fluids exhibiting a shear-thinning property (a pseudoplasticity behavior).
Same trends of K value and hysteresis ring area of starch added with polysaccharides have been reported before, and these results indicated that the relative stability and uniformity of the starch gel structure was enhanced
The value of n is inversely correlated with the pseudoplasticity degree of the entire system , PPC replacement decreasing the n value, for PPC1 the most. It meant that the presence of PPC made the entangled starch molecular chains have a greater tendency to be straightened or dispersed under the action of external force, which was conducive to the flow of starch paste, consequently lowering the viscosity more obviously. The result was unanimous to the change of RVA test
3.3.2. Dynamic rheological analysis
Starch-PPC pastes had better gelling properties than starch paste, and the lower the degree of PPC polymerization, the more obvious the effect. PPC addition could make PoS more suitable candidates as gelling agents for food industry manufacture including soft candies, ice cream and meat foods production, thereby improving these products' nutrition, texture and stability
3.4. Thermodynamic properties
The addition of PPC increased the gelatinization transition temperatures (To, Tp and Tc) of PoS within 2 °C, indicating PPC could delay starch gelatinization. The results were supposedly attributed to the partial undissolved PPC attached to the starch granules, which enlarged steric hindrance and suppressed the swelling and fracture of starch
The OH groups of PPC interacted with the side chain of amylopectin and bound to the amorphous region of starch granules to varying degrees. It facilitated the transition from the water-absorbing swelling in the amorphous zone to the crystalline zone for the starch granules, thereby altering the coupling force between the crystalline and amor-phous matrix
Among the three types of PPC, PPC3 had the strongest suppression effect on starch retrogradation, then PPC2 and finally PPC1
3.5. X-ray diffraction patterns
Studies have shown that starch will form V-type crystals when combined with fatty acids, iodine and other substances, and the V-type inclusion shows characteristic diffraction peaks at 20.0° and sometimes at 7.0° and 13.0°. The complex had obvious diffraction peak at 2θ = 13.1°, which is mainly due to the formation of amylose and lipid complexes, further hindering amylose rearrangement
Long-chain PPC affected the overall distribution and arrangement of the starch chains rather than short-range double helix structure of the starch. This result was consistent with the DSC.
Through our research, we found that PPC as a functional compound can not only serve as a mechanical reinforcement to form interpenetrating network with starch and enhance the mechanical properties of starch films, but also suppress the recrystallization and retrogradation of starch, thus maintaining the preserved starch film flexible and stretchable
3.6. In vitro digestibility
Based on these findings by adding candidate PPC, it is expected to produce natural anti-diabetic drugs or chronically digested starch that is beneficial for postprandial blood glucose control
Q6. 4. Conclusions
The authors used in this study three types of PPC with different DPs (6.39 ± 0.13, 8.21 ± 0.76 and 9.92 ± 0.21, respectively) to investigate the effects of PPC on the physicochemical and in vitro digestion properties for better application of PPC in starchy food
PPC enhanced the thermal stability and delayed starch gelatinization of PoS. It pronouncedly affected the elastic properties more than the viscous properties of this pasting
Starch digestion and retrogradation could be restrained by PPC with different DPs in varying degrees. Among the three, PPC with lower DP more evidently influenced starch gelling performance, whereas larger molecules exhibited a greater impact on starch long-term retrogradation and digestive properties
It might be attributed to the different type of starch chains combined with PPC of different DP. The results of current study may render a beneficial recommendation for the use of PPC to modify pasting, thermal, rheological and digestive properties of PoS-based foods.
Q7. References
Are good enough in total number (619 well selected, and from recent years)

Author Response
Response to Reviewer3 Comments
Q1: Title: Must be shorten and suppress polymeric at least and can be added “for three types of polymeric and resumed as follow: “Effects of three types of polymeric proanthocyanidin on physiochemical …..
Response 1: Thank you very much for your good suggestion. As suggested, the title “Effects of polymeric proanthocyanidin with different degrees of polymerization on physicochemical and in vitro digestive properties of potato starch” was changed to “Effects of three types of polymeric proanthocyanidins on physicochemical and in vitro digestive properties of potato starch” (Please see page 1, lines 2, 3 in the revised manuscript)
Q2 & Q3 & Q4 & Q5 & Q6 & Q7: Abstract and Keywords, Introduction, Materials and Methods, Results and Discussion, Conclusions, References
Response 2: Thank you very much for your good suggestion. We have checked these sections, and we thank your positive comments on our manuscript.
Thank you again. We hope that you could be satisfied with our changes.

Reviewer 4 Report
Comments to authors
The publication is interesting but needs a few corrections.
Detailed comments:
- Materials and Methods section:
Rapid viscosity analysis (RVA) section needs more detailed description, temperature range in particular.
- Results and discussion section
The table 2 should be moved and Figure 2 maybe too.
- References
Item 1: The name of the journal is missing.
The manner of stating the name of journals should be standardized. Nowadays there is sometimes a full name, sometimes abbreviation.
Author Response
Response to Reviewer4 Comments
The publication is interesting but needs a few corrections.
Point 1: Detailed comments: Materials and Methods section:
Rapid viscosity analysis (RVA) section needs more detailed description, temperature range in particular.
Response 1: Thank you very much for your good suggestion. We've described it in detail (Please see page 3, lines 111-116 in the revised manuscript).
Point 2: Results and discussion section
The table 2 should be moved and Figure 2 maybe too.
Response 2: Thank you very much for your good suggestion. We have swapped the positions of Figure 2 and Table 2 (Please see page 7, (Figure 2, Table 2) in the revised manuscript).
Point 3: References
Item 1: The name of the journal is missing.
The manner of stating the name of journals should be standardized. Nowadays there is sometimes a full name, sometimes abbreviation.
Response 3: Thank you for your good suggestions. We have added the missing name of the journal and standardized the name of journals by using abbreviations in References (Please see pages 12-15 in the revised manuscript).
Thank you again. We hope that you could be satisfied with our changes.
